# Physical Characteristics and Body Image of Japanese Female University Long-Distance Runners



**Masaharu Kagawa** [1,2,3,4,5,*], **Sayumi Iwamoto** [6,7], **Kazuko Ishikawa-Takata** [8] **and Masako Ota** [6]

1   Institute of Nutrition Sciences, Kagawa Nutrition University, Saitama 350-0288, Japan
2   School of Population Health, Curtin University, Perth, WA 6845, Australia
3   School of Exercise and Nutrition Sciences, Queensland University of Technology, Brisbane, QLD 4059, Australia
4   Faculty of Public Health, Mahidol University, Bangkok 10400, Thailand
5   Faculty of Public Health, Universitas Airlangga, Surabaya 60115, Indonesia
6   Faculty of Health and Sports Sciences, Toyo University, Tokyo 115-8650, Japan; siwamoto@toyo.jp (S.I.); masako@toyo.jp (M.O.)
7   Sports Performance Research Institute New Zealand, Auckland University of Technology, Auckland 1010, New Zealand
8   Faculty of Applied Biosciences, Tokyo University of Agriculture, Tokyo 156-8502, Japan; kt207460@nodai.ac.jp
*   Correspondence: mskagawa@eiyo.ac.jp; Tel.: +81-49-281-7743

**Featured Application: Many Japanese female university long-distance runners misperceive their current weight and adiposity. However, their body dissatisfaction may not be based on their physique or external information sources. Understanding their past experience and its impact on their body image may be beneficial. Further research on the effects of educational programs for the improvement of performance on the correction of body image and behaviour modification may be warranted.**

**Abstract:** While female long-distance runners are considered to have strong body dissatisfaction and body concerns, body-image research that incorporates detailed anthropometric and body composition parameters is still limited. The present study therefore investigates the physical characteristics and body image of Japanese female long-distance runners and explores the factors that influence their body image. Detailed anthropometric and body composition assessment using a dual-frequency bio-electrical impedance analysis (DFBIA) was conducted on 30 Japanese female university long-distance runners. In addition, a questionnaire that included the Body Satisfaction Scale (BSS) and the Ben-Tovim Walker Body Attitudes Questionnaire (BAQ) was administered. On average, the participants had relatively low body mass index (BMI) and percentage body fat (%BF) (BMI: $18.3 \pm 1.6$ kg/m$^2$; %BF: $19.7 \pm 4.4$%), but about 50–60% of them perceived themselves as being fat or having an excessive level of %BF. Their BSS scores were not associated with their measured physique. However, the anthropometric variables of the limbs were associated with the BAQ and its subscales. There was no single source that the majority referred to obtain information on their body, and performance was the only reason for their increased body concern. In order to better understand the factors that influence their body dissatisfaction and the effects of providing accurate information on behaviour modification, further investigation is warranted.

**Keywords:** females; long-distance runners; athletes; Japanese; body image; anthropometry; body composition

## 1. Introduction

Sports performance is known to be associated with specific types of physiques. An ideal physique with an appropriate body size, proportion and body composition varies depending on the sporting event. Some sports have shown changes in the parameters of

body size (e.g., stature and body mass) over the past decades [1,2]. In track and field events, the body size of runners participating in short-distance sprint events (e.g., 100 m and 200 m) as determined using body mass index (BMI) has increased throughout the 20th century, while that of long-distance runners, particularly in marathons, has decreased in the same period of time [1]. Such differences in a secular trend of body size for different sporting events are because body size associates with the performance of athletes. In running events, BMI has been reported to be associated with running speed [3]. In addition, although further evidence is required, a systematic review suggested that a higher BMI may be one of the potential risk factors for overuse leg injuries [4]. However, for long-distance runners, controversy remains regarding the association between the BMI and performance of runners, as a number of studies reported no correlations between body mass or BMI and performance [5,6].

While the associations between body mass or BMI and performance have not been confirmed in long-distance running to date, female runners in particular are reported to have a lower BMI [6,7] than male runners [3,5,6,8] and often have BMIs that are considered "underweight" (i.e., below 18.5 kg/m$^2$) [9,10] according to the classification of the World Health Organization (WHO) [11]. Excessive concern about weight and preoccupation with weight loss among female long-distance runners may be associated with distorted body image. Body image can be defined as any image or values of the human body—which may concern spatial existence; motion that the body causes; visual shape, size, and/or composition, either as a whole or as part—that is constructed and altered throughout the lifetime [12]. Individuals with distorted body image are considered to be associated with unnecessary weight-loss practices, including skipping breakfast, overtraining and the use of diuretics, and may also have an increased risk of developing eating disorders. A number of previous studies reported that BMI is associated with body dissatisfaction and the risk of disordered eating behaviours among female long-distance runners [13–15]. Considering a study that reported no observation of such an association in males [14], it may be suggested that female long-distance runners are more vulnerable to having distorted body image than males.

While many studies have focused on associations of body-image parameters with weight or BMI, studies that have incorporated other physical characteristics such as skin-folds and percentage body fat (%BF) are scarce. Because low skeletal muscle mass or fat-free mass (FFM) affects the running performance of athletes, it is important to investigate if athletes acknowledge and take their body composition into account when they justify their physical status and therefore their construction of their body image. Many body-image studies on long-distance runners have been based on Caucasians, and, to the best of our knowledge, no studies have examined the body image of Japanese or Asian female long-distance runners. Asians including Japanese are known to have a higher %BF at a lower BMI than Caucasians [16–18]. Moreover, Japanese females have poor body perception of their own adiposity compared to their male counterparts [19]. These studies suggest that Japanese female long-distance runners' excessive preoccupation with weight alone may result in a misinterpretation of their actual body composition, especially regarding the amount of skeletal muscle mass, which is an essential component for sport performance, and limiting their weight gain may result in the disturbance of both skeletal muscle mass development and improvements to their performance. However, body-image research that incorporates detailed anthropometric and body composition parameters is lacking in Japan, particularly on athletes. The present study, therefore, aims to (1) investigate the anthropometric and body composition characteristics of Japanese female long-distance runners and (2) explore their body image, namely, body perception, body satisfaction and body concern, together with the risk of disordered eating behaviours. Furthermore, this study aims to (3) determine the factors that influence their body image.

## 2. Materials and Methods

### *2.1. Participants*

Female university long-distance runners participating in an *Ekiden* (Japanese long-distance relay race) event were invited to the study. A total of 31 participants were measured during the period between December 2017 and May 2019 as a part of a longitudinal project. In the present study, the baseline data of the longitudinal dataset were analysed. All participants lived in the same dormitory provided by the university, and both breakfast and dinner were provided by the team dietitian on weekdays. Verbal and written information on the research project was provided to all participants, including information on the voluntary nature of their participation, confirmation that there would be no adverse consequences for their withdrawal and a privacy policy as well as data storage and elimination policies, prior to their participation. Those who understood the nature of the study and agreed to participate were instructed to sign the written informed consent form. The study was conducted in accordance with the Declaration of Helsinki (2013 revised edition, Fortaleza) and was approved by the Ethics Committee of Toyo University (TU2017-010TU2017-H0052018-H002).

### *2.2. Methods*

All participants were instructed to complete a (1) self-administered questionnaire including validated body-image scales, (2) detailed anthropometry and (3) body composition assessment.

### 2.2.1. Self-Administered Questionnaire

A questionnaire booklet with questions on demographics and body image was distributed to each participant. Items on body image included (a) ideal height, (b) ideal weight for current height, (c) ideal weight for ideal height, (d) ideal weight for the competition, (e) perceived physique, (f) perceived body fatness and (g) perceived anxiety about loss of muscle mass. Participants were also asked about the major sources of information that influence their ideal physique and body composition and reasons to be concerned about their own physique. Participants selected all that applied from the choices prepared. In addition to the above-mentioned questions, validated scales, namely, the Body Satisfaction Scale (BSS), the Ben-Tovim Walker Body Attitudes Questionnaire (BAQ) and the 26-item Eating Attitudes Test (EAT-26), were also administered. Participants were instructed to complete the questions prior to anthropometry and body composition assessments to avoid bias. The booklet was collected and checked by the researcher in a face-to-face setting for any missing information and for confirmation of responses.

### The Body Satisfaction Scale (BSS)

The BSS was developed to assess body satisfaction/dissatisfaction based on assessments of 16 body parts [20]. Participants were instructed to rate each body part using a 7-point Likert scale that ranged from (1) very satisfied to (7) very unsatisfied. The scale can present body satisfaction with each body part, the whole body ($BSS_{General}$), the head region ($BSS_{Head}$) and the trunk and limb regions ($BSS_{Body}$). While Slade [20] reported that "ears" and "neck" can be omitted from the calculation of $BSS_{Head}$ and $BSS_{Body}$ scores, the present study included both variables to follow the same protocol of a Japanese-translated version [21]. The higher the score, the greater the dissatisfaction. For the present study, a Japanese-translated version [21] was utilized, and the Cronbach's alpha was 0.924.

### The Ben-Tovim Walker Body Attitudes Questionnaire (BAQ)

The BAQ is a 44-item self-administered questionnaire designed to assess attitudes which individuals hold on their own bodies [22]. Participants were instructed to rate each question using a 5-point Likert scale that ranges from (1) strongly disagree to (5) strongly agree. From the assessment, the total score ($BAQ_{Total}$) as well as scores for the following subscales were determined: (1) feeling fat ($BAQ_{Fat}$), (2) body disparagement ($BAQ_{Disparagement}$),

(3) strength and fitness (BAQ$_{Strength}$), (4) salience of weight and shape (BAQ$_{Salience}$), (5) attractiveness (BAQ$_{Attractive}$) and (6) lower body fatness (BAQ$_{Lowerbody}$). The present study used a Japanese-translated version [23]. The Cronbach's alpha obtained from the study group was 0.879.

The 26-Item Eating Attitudes Test (EAT-26)

The EAT-26 is a scale commonly utilized to examine the risk of disordered eating behaviours, including eating disorders [24]. In addition to the total score (EAT$_{Total}$), the EAT-26 has three subscales: (1) dieting (EAT$_{Diet}$), (2) bulimia and food preoccupation (EAT$_{Bulimia}$) and (3) oral control (EAT$_{Oral}$). Participants were instructed to rate each question using a 6-point Likert scale that ranges from (1) never to (6) always. In order to avoid a skewed distribution of the results that may be obtained from a group of non-eating disordered participants, the current study adopted the 6-point scoring method proposed by Wells and colleagues [25]. A Japanese-translated version by Mukai and colleagues [26] was used in the present study. The Cronbach's alpha obtained from the study group was 0.915.

### 2.2.2. Anthropometry

All participants were assessed using a detailed anthropometric protocol consisting of 22 variables (stature, body mass, 8 skinfolds, 6 girths, 2 lengths and 4 breadths). All measurements were conducted according to the protocol of the International Society for the Advancement of Kinanthropometry (ISAK) [27]. Stature was measured using a stadiometer (GPM, Seritex Inc., New York, NY, USA) to the nearest 0.1 cm, and body mass was measured to the nearest 0.1 kg using a dual-frequency bioelectrical impedance analysis (DFBIA) device (Innerscan Dual RD-800, Tanita Corp., Tokyo, Japan). Skinfolds were measured to the nearest 0.1 mm using a Harpenden skinfold calliper (Baty International Ltd., Burgess Hill, UK). Girths were measured using a steel tape measure (W606PM, Apex Tool Group LLC, Apex, NC, USA), whereas bone lengths and breadths were measured to the nearest 0.1 cm using a segmometer, a large sliding calliper and a small sliding calliper (Rosscraft Innovations Inc., Surrey, BC, Canada). All measurements were taken by anthropometrists accredited by ISAK with acceptable levels of intra- and inter-tester technical error of measurements (TEMs) [28,29]. From the measurements, BMI, sum of eight skinfolds ($\Sigma$8SF), ratio score (body mass [kg]/ $\Sigma$8SF [mm]), leg length and leg length relative to stature were calculated. To determine the distribution of body size, the BMIs of participants were subdivided according to the WHO and the Japan Society for the Study of Obesity (JASSO) cut-off points as well as the public health action points considered appropriate to reflect body fat accumulation for Asians (i.e., <18.5 kg/m$^2$, 18.5–22.9 kg/m$^2$, 23.0–24.9 kg/m$^2$ and $\geq$25.0 kg/m$^2$) [11,16,30].

### 2.2.3. Body Composition Assessment

Body composition was assessed using a DFBIA device (Innerscan Dual RD-800, Tanita Corp., Tokyo, Japan) [31]. After removing all jewellery and metal and then wiping the surface of both hands and feet, participants were instructed to stand on the device and maintain an upright posture. From the measurement, percentage body fat (%BF), fat-free soft tissue mass, bone mass and proportion of total body water were determined. From the obtained variables, fat mass (FM: %BF/100 × body mass), fat-free mass (FFM: Body mass—FM), fat mass index (FMI: FM/stature$^2$) and fat-free mass index (FFMI: FFM/stature$^2$) were calculated. In addition, abdominal fat was examined using a BIA device particularly designed to estimate percentage abdominal fat (%AF) (AB-140, Tanita Corp., Tokyo, Japan).

### 2.3. Statistical Analysis

Of the 31 participants who underwent assessments, 1 participant has been excluded because of a difference in the event that she is an expert in (race walking), resulting in the inclusion of 30 participants. Normality of data was examined using the Shapiro–Wilk test and normal Q–Q plots. Variables with confirmed normality are expressed as

mean $\pm$ standard deviation, while those that did not show normality are expressed as median and quartiles. Measured stature was compared with reported "ideal" height to calculate difference between current and ideal values (Difference$_{IdealHt}$). Similarly, difference between measured body mass and reported ideal weight (Difference$_{IdealWt}$) was calculated as well as difference between current and ideal weight for the current height (Difference$_{CurrentHt}$) and difference between current and perceived ideal weight for the competition (Difference$_{Competition}$). Furthermore, BMI was calculated on the basis of (1) measured stature and body mass (BMI), (2) perceived ideal height and weight (BMI$_{Ideal}$), (3) measured stature and perceived ideal weight for the current height (BMI$_{IdealWt}$) and (4) measured stature and perceived ideal weight for the competition (BMI$_{Competition}$). The above-mentioned differences between measured and perceived values and calculated BMIs were compared using paired t-tests. Chi-squared tests were conducted to assess associations between current BMI and perceived physique as well as body fatness. In order to examine associations between anthropometric and body composition variables on body-image scales (i.e., BSS, BAQ, EAT-26 and their subscales), stepwise regression analyses were conducted. All statistical analyses were conducted using the IBM SPSS Statistics package (version 27.0, IBM, Tokyo, Japan). All statistical tests used a significance level of 0.05 unless otherwise stated.

## 3. Results

### 3.1. Anthropometric and Body Composition Profiles

Participants had a median age of 19.0 years with an average stature, body mass and BMI of 158.4 $\pm$ 6.0 cm, 46.0 $\pm$ 5.1 kg and 18.3 $\pm$ 1.6 kg/m$^2$, respectively (Table 1). On average, participants had $\Sigma$8SF of 64.7 $\pm$ 17.0 mm with the greatest subcutaneous fat accumulation around the thigh region. The participants had a median value of 0.70 for the ratio score and had a mean somatotype of balanced ectomorph (2.4-2.8-3.8).

**Table 1.** Anthropometric profile of the participants.

| Variables | | Participants (n = 30) | |
|---|---|---|---|
| | | **Mean $\pm$ SD (Median; Quartile)** | **Min–Max** |
| | Age (years) * | (19.0; 18.0–20.0) | 18.0–21.0 |
| | Stature (cm) | 158.4 $\pm$ 6.0 | 148.3–171.7 |
| | Body mass (kg) | 46.0 $\pm$ 5.1 | 36.1–56.0 |
| Skinfolds (mm) | Triceps | 10.5 $\pm$ 2.8 | 3.1–16.7 |
| | Subscapular * | (6.4; 5.7–7.5) | 4.1–11.5 |
| | Biceps | 4.3 $\pm$ 1.5 | 2.0–9.0 |
| | Iliac crest * | (8.7; 6.8–11.5) | 3.6–22.2 |
| | Supraspinale | 4.9 $\pm$ 1.3 | 2.5–8.0 |
| | Abdominal * | (6.8; 5.5–8.5) | 3.8–19.6 |
| | Front thigh | 15.1 $\pm$ 4.2 | 4.8–21.1 |
| | Medial calf | 6.1 $\pm$ 2.2 | 2.0–11.1 |
| Girths (cm) | Arm relaxed | 22.1 $\pm$ 1.7 | 18.0–24.9 |
| | Arm flexed and tensed | 22.7 $\pm$ 1.5 | 19.4–25.3 |
| | Waist | 61.9 $\pm$ 3.4 | 54.6–70.1 |
| | Gluteal | 84.4 $\pm$ 4.1 | 75.0–91.5 |
| | Mid-thigh * | (46.1; 43.9–47.8) | 36.7–50.5 |
| | Calf maximum | 33.4 $\pm$ 1.9 | 28.5–38.9 |
| Lengths (cm) | Tronchanterion-tibiale laterale | 39.9 $\pm$ 2.2 | 36.3–44.0 |
| | Tibiale laterale ht | 40.5 $\pm$ 2.2 | 37.4–45.4 |

**Table 1.** *Cont.*

| Variables | | Participants (n = 30) | |
| --- | --- | --- | --- |
| | | Mean $\pm$ SD (Median; Quartile) | Min–Max |
| Breadths (cm) | Biacromial | 35.0 $\pm$ 1.6 | 31.9–38.5 |
| | Biiliocristal * | (26.0; 25.2–26.8) | 22.3–30.3 |
| | Biepicondylar humerus | 5.5 $\pm$ 0.2 | 5.2–6.0 |
| | Biepicondylar femur | 8.3 $\pm$ 0.4 | 7.7–9.3 |
| Anthropometric indices | Body mass index (kg/m$^2$) | 18.3 $\pm$ 1.6 | 14.6–20.7 |
| | Sum of 8 skinfolds (mm) | 64.7 $\pm$ 17.0 | 27.4–105.5 |
| | Ratio score (kg/mm) * | (0.70; 0.63–0.85) | 0.42–1.34 |
| | Leg length (cm) | 80.4 $\pm$ 4.0 | 73.7–88.0 |
| | Percentage leg length (%) | 50.7 $\pm$ 1.0 | 48.5–52.8 |
| Somatotype | Endomorphy | 2.4 $\pm$ 0.7 | 0.8–4.1 |
| | Mesomorphy | 2.8 $\pm$ 0.9 | 0.6–4.7 |
| | Ectomorphy | 3.8 $\pm$ 1.1 | 2.1–6.4 |

\* Results are presented as median, 25th and 75th quartiles. SD: standard deviation.

The results from the body composition assessments are presented in Table 2. The participants had an average %BF of 19.7 $\pm$ 4.4% with an FM and FFM of 9.2 $\pm$ 2.7 kg and 36.8 $\pm$ 3.3 kg, respectively. From the abdominal fat assessment, it was estimated that the participants had a mean %AF of 18.4 $\pm$ 2.9%. FMI and FFMI were 3.7 $\pm$ 1.0 kg/m$^2$ and 14.7 $\pm$ 0.7 kg/m$^2$, respectively.

**Table 2.** Body composition of Japanese female long-distance runners.

| Variables | Participants (n = 30) | |
| --- | --- | --- |
| | Mean $\pm$ SD | Min–Max |
| Percentage body fat (%) | 19.7 $\pm$ 4.4 | 8.3–27.7 |
| Lean soft tissue mass (kg) | 34.6 $\pm$ 2.8 | 28.9–41.2 |
| Bone mass (kg) | 2.0 $\pm$ 0.3 | 1.1–2.7 |
| Percentage total body water (%) | 54.6 $\pm$ 2.6 | 50.2–61.0 |
| Fat mass (kg) | 9.2 $\pm$ 2.7 | 3.0–14.4 |
| Fat-free mass (kg) | 36.8 $\pm$ 3.3 | 30.4–43.9 |
| Percentage abdominal fat (%) | 18.4 $\pm$ 2.9 | 11.8–22.2 |
| Fat mass index (kg/m$^2$) | 3.7 $\pm$ 1.0 | 1.2–5.7 |
| Fat-free mass index (kg/m$^2$) | 14.7 $\pm$ 0.7 | 13.1–16.2 |

SD: standard deviation.

### 3.2. Body Perception, Satisfaction and Concern of the Participants

Table 3 presents the responses to the body-image questions and scales. Compared with the measured values, the participants wished to be 1.2 cm taller ($p < 0.05$) and from 2.3 to 2.7 kg lighter ($p < 0.01$), depending on their current height. The participants also wished to have a 2.7 kg lighter weight on average for better performance at competitions ($p < 0.01$). Subsequently, BMI$_{Ideal}$, BMI$_{IdealWt}$ and BMI$_{Competition}$ were considerably lower (ranging between 16.6 and 17.2 kg/m$^2$) than the BMI calculated from measured stature and body mass. While most participants (86.4%) reported that they put in effort to maintain or achieve their perceived ideal weight, interestingly, only 45.4% reported a past experience of weight-loss behaviours. On the other hand, a vast majority (95.5%) eat three meals a day and are aware of the contents and amounts of their meals.

**Table 3.** Body image of Japanese female long-distance runners.

| Variables | | Results |
|---|---|---|
| Height (cm) | Ideal | 159.6 ± 4.4 |
| | Difference$_{IdealHt}$ | −1.2 ± 3.2 * |
| Weight (kg) | Ideal [†] | 43.9 ± 4.0 |
| | Ideal for current height [†] | 43.1 ± 4.4 |
| | For competition | 43.3 ± 4.0 |
| | Difference$_{IdealWt}$ [†] | 2.3 ± 2.8 ** |
| | Difference$_{CurrentHt}$ [†] | 2.7 ± 2.8 ** |
| | Difference$_{Competition}$ | 2.7 ± 2.6 ** |
| BMI (kg/m$^2$) | BMI$_{Ideal}$ [†] | 17.2 ± 1.2 ** |
| | BMI$_{IdealWt}$ [†] | 16.6 ± 3.3 * |
| | BMI$_{Competition}$ | 17.2 ± 1.0 ** |
| Effort to maintain current weight or to achieve ideal weight [‡] | % Yes | 86.4 |
| Experience of weight-loss behaviours | % Yes | 45.4 |
| Perceived physique [‡] (%) | Too thin | 4.5 |
| | Just right | 27.3 |
| | Slightly fat | 50.0 |
| | Too fat | 4.5 |
| | Not sure | 13.6 |
| Perceived body fat [‡] (%) | Too small | 4.5 |
| | Just right | 13.6 |
| | Slightly large amount | 40.9 |
| | Too much | 22.7 |
| | Not sure | 18.2 |
| Perceived anxiety about muscle loss [‡] (%) | Not anxious at all | 4.5 |
| | Almost no anxiety | 9.1 |
| | Do not feel anything | 9.1 |
| | Slightly anxious | 63.6 |
| | Very anxious | 13.6 |
| BSS [‡] | BSS$_{General}$ | (68.0; 63.5–71.3) |
| | BSS$_{Head}$ | (33.5; 31.8–35.0) |
| | BSS$_{Body}$ | (34.0; 32.8–37.3) |
| BAQ [‡] | BAQ$_{Total}$ | 130.0 ± 17.9 |
| | BAQ$_{Fat}$ | 41.0 ± 9.5 |
| | BAQ$_{Disparagement}$ | (17.5; 15.0–21.3) |
| | BAQ$_{Strength}$ | 19.1 ± 2.6 |
| | BAQ$_{Salience}$ | 23.6 ± 5.3 |
| | BAQ$_{Attractiveness}$ | 13.9 ± 2.7 |
| | BAQ$_{Lowerfat}$ | 13.9 ± 3.3 |
| EAT-26 [#] | EAT$_{Total}$ | 59.3 ± 18.5 |
| | EAT$_{Diet}$ | 31.1 ± 10.3 |
| | EAT$_{Bulimia}$ | (12.0; 7.0–16.0) |
| | EAT$_{Oral}$ | 16.2 ± 6.2 |

Results are expressed as mean ± standard deviation or (median; quartiles) or percentage (%); [†] n = 29; [‡] n = 22; [#] n = 18; * $p < 0.05$ when compared with measured values using paired *t*-test; ** $p < 0.01$ when compared with measured values using paired *t*-test. Difference$_{IdealHt}$: difference between measured stature and reported ideal height; Difference$_{IdealWt}$: difference between measured body mass and reported ideal weight; Difference$_{CurrentHt}$: difference between measured body mass and reported ideal weight for current height; Difference$_{Competition}$: difference between measured body mass and reported ideal weight for the competition; BMI$_{Ideal}$: BMI calculated from perceived ideal height and weight; BMI$_{IdealWt}$: BMI calculated from measured stature and perceived ideal weight for current height; BMI$_{Competition}$: BMI calculated from measured stature and perceived ideal weight for the competition; BSS: Body Satisfaction Scale; BAQ: Ben-Tovim Walker Body Attitudes Questionnaire; EAT-26: Eating Attitudes Test-26; BSS$_{General}$: BSS general score; BSS$_{Head}$: BSS head subscale score; BSS$_{Body}$: BSS body subscale score; BAQ$_{Total}$: BAQ total score; BAQ$_{Fat}$: BAQ feeling fat subscale score; BAQ$_{Disparagement}$: BAQ disparagement subscale score; BAQ$_{Strength}$: BAQ strength subscale score; BAQ$_{Salience}$: BAQ salience subscale score; BAQ$_{Attractiveness}$: BAQ attractive subscale score; BAQ$_{Lowerfat}$: BAQ lower body fat subscale score; EAT$_{Total}$: EAT total score; EAT$_{Diet}$: EAT diet subscale score; EAT$_{Bulimia}$: EAT bulimia subscale score; EAT$_{Oral}$: EAT oral control subscale score.

Of those who responded to the questions, only 27.3% and 13.6% of the participants perceived that they have about the right physique and %BF, respectively. In comparison, 54.5% perceived themselves as being fat, and 63.6% perceived themselves as having an

excessive amount of body fat. In addition, 13.6–18.2% were unable to justify their current physique and/or %BF level. There were no significant differences in the perceptions of their physique and adiposity between participants of different BMI or %BF levels, respectively. In relation to muscle tissues, more than 75% expressed anxiety about the loss of muscle tissues.

The present study incorporated multiple scales in order to understand the level of body dissatisfaction, body concerns and the risk of disordered eating behaviours. The median of the $BSS_{General}$ score was 68 with equivalent scores obtained from the $BSS_{Head}$ and the $BSS_{Body}$ subscales. The mean values obtained from the $BAQ_{Total}$ and the $EAT_{Total}$ were $130.0 \pm 17.9$ and $31.1 \pm 10.3$, respectively.

### 3.3. Factors Influencing the Body Image of Female Long-Distance Runners

In order to examine the associations between the anthropometric and body composition variables and body-image scales, stepwise regression analyses were conducted (Table 4). Among the scales and their subscales examined, $BSS_{General}$ and its subscales (i.e., $BSS_{Head}$ and $BSS_{Body}$) and $BAQ_{Disparagement}$, $EAT_{Total}$ and the $EAT_{Bulimia}$ subscale did not show associations with the anthropometric and body composition variables. On the other hand, biceps skinfold was found to be an important anthropometric variable for $BAQ_{Total}$ and for three BAQ subscales (i.e., $BAQ_{Fat}$, $BAQ_{Salience}$ and $BAQ_{Lowerbody}$). Among the associated anthropometric parameters, those measured from the limbs, including triceps, biceps, front thigh skinfolds and mid-thigh and calf maximum girths, were found to have an important and strong influence on BAQ. Compared with the BAQ scores that showed associations with multiple anthropometric variables, the $EAT_{Diet}$ and $EAT_{Oral}$ subscales were found to be associated with only one anthropometric variable each with a lower $R^2_{Adj}$.

**Table 4.** Regression equations for scale scores and anthropometric variables.

| Dependent Variables | Variables | Standardized Coefficient β | *p*-Value | $R^2_{Adj}$ |
|---|---|---|---|---|
| $BAQ_{Total}$ | Biceps skinfold | 0.653 | 0.002 | 0.679 |
| | Age | −0.579 | <0.001 | |
| | Calf maximum girth | −0.625 | 0.002 | |
| | Triceps skinfold | 0.610 | 0.008 | |
| $BAQ_{Fat}$ | Biceps skinfold | 0.609 | 0.003 | 0.340 |
| $BAQ_{Strength}$ | Subscapular skinfold | 0.918 | <0.001 | 0.661 |
| | Triceps skinfold | −1.189 | <0.001 | |
| | Front thigh skinfold | 0.605 | 0.024 | |
| $BAQ_{Salience}$ | Biceps skinfold | 0.809 | 0.002 | 0.345 |
| | Calf maximum girth | −0.559 | 0.022 | |
| $BAQ_{Attractiveness}$ | Mid-thigh girth | −0.740 | 0.002 | 0.516 |
| | Body mass | 0.930 | <0.001 | |
| | Arm flexed and tensed girth | −0.575 | 0.017 | |
| $BAQ_{Lowerbody}$ | Biceps skinfold | 0.598 | <0.001 | 0.583 |
| | Humerus breadth | −0.384 | 0.017 | |
| | Age | −0.329 | 0.039 | |
| $EAT_{Diet}$ | Triceps skinfold | 0.522 | 0.018 | 0.232 |
| $EAT_{Oral}$ | Arm flexed and tensed girth | −0.603 | 0.005 | 0.329 |

Anthropometric variables with significant associations with body-image scale scores are presented. $BAQ_{Total}$: Ben-Tovim Walker Body Attitudes Questionnaire (BAQ) total score; $BAQ_{Fat}$: BAQ feeling fat subscale; $BAQ_{Strength}$: BAQ strength and fitness subscale; $BAQ_{Salience}$: BAQ salience of weight and shape subscale; $BAQ_{Attractiveness}$: BAQ attractiveness subscale; $BAQ_{Lowerbody}$: BAQ lower body fat subscale; $EAT_{Diet}$: 26-item Eating Attitudes Test (EAT) diet subscale; $EAT_{Oral}$: EAT oral control subscale; $R^2_{Adj}$: adjusted coefficient of determination.

In addition to the physical parameters that influence the validated scale scores, major sources of information on physique and body composition were examined (Figure 1). While the vast majority did not utilize any sources to gain information on physique and body

composition, 40.9% obtained information from the Internet, followed by TV (27.3%) and friends (22.7%).

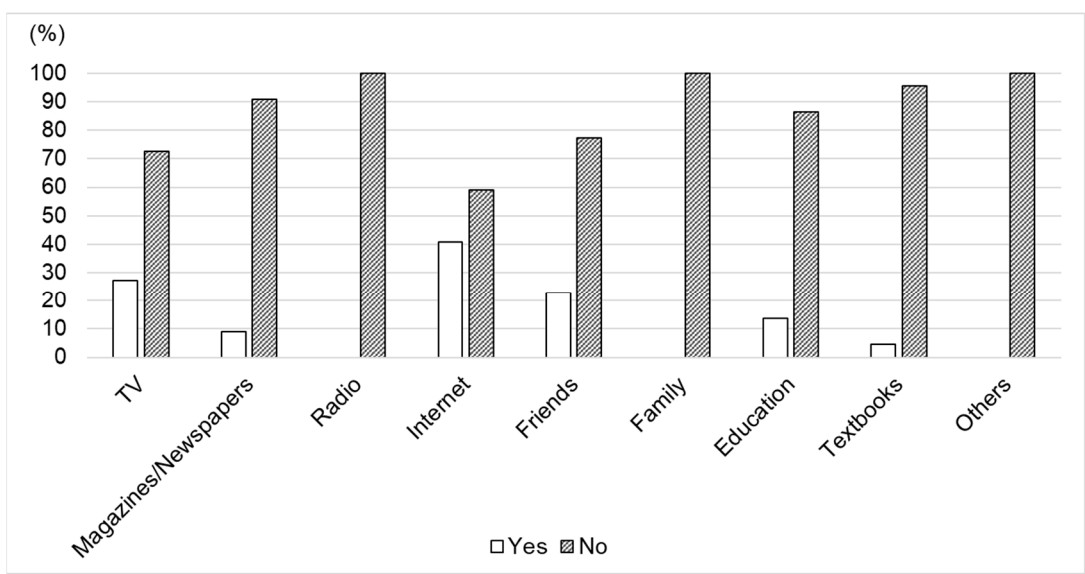

**Figure 1.** Sources that influence participants' ideal physique and body composition (n = 22).

In addition to the information sources, the reasons for being concerned about their own physique were questioned (Figure 2). Almost all participants reported their performance as the major reason (95.5%). In addition, appearance, to gain relief and/or confidence and being surrounded by slim people were equally expressed as reasons for being concerned about their physiques (36.4%).

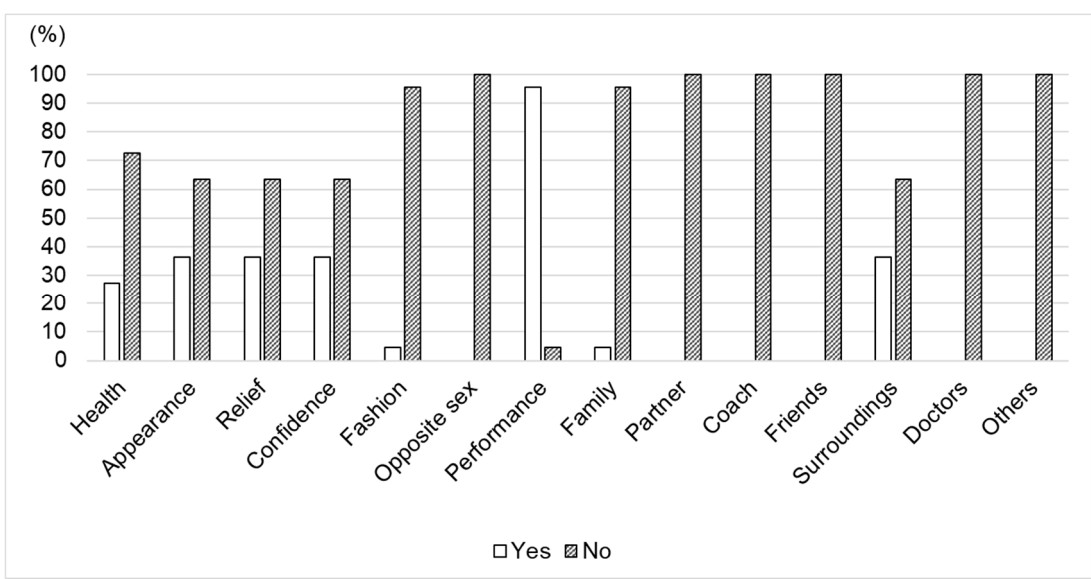

**Figure 2.** Reasons to be concerned about own physique (n = 22).

## 4. Discussion

The present study investigated the physical characteristics and body-image parameters of Japanese female university long-distance runners. The body sizes of the participants as measured with stature and body mass were taller and lighter than those of the comparable age group as reported in a recent National Health and Nutrition Survey in Japan (n = 15, 156.7 ± 7.4 cm and 48.7 ± 7.5 kg) [32]. The observed results are in line with

previous studies that compared female long-distance runners and non-athletes [9,33]. In addition, while the present study assessed university athletes, their body sizes and calculated BMIs were comparable to those of elite long-distance runners of different ethnic backgrounds [6,9,10]. While a relatively small sample size contributed to the results, consistent results with previous studies indicate the possibility of increasing body size among Japanese female university long-distance runners over the last few decades. This may be a result of morphological optimization that may be achieved using talent identification and selection processes.

In addition to body size, the present study examined subcutaneous fat accumulation according to anthropometry and body composition using DFBIA. The results show that the participants had sums of skinfolds that appear comparable to those found in a previous study on middle- and long-distance runners of different ethnic backgrounds [34]. However, considering the body size differences between groups, the results may indicate greater subcutaneous fat accumulation among Japanese female university long-distance runners. The body composition of female long-distance runners has been reported in a number of studies to date. However, the results vary between studies ($11.4 \pm 3.6\%$ to $24.1 \pm 4.7\%$) [7,9,33,35,36]. This variance may be partly due to the characteristics of the participants and the body composition assessment methods used. The results from the current study ($19.7 \pm 4.4\%$) fall within the range of previous studies and may be considered as reflecting a relatively low %BF for females. The average physique of the participants with a low BMI (i.e., a light body mass relative to their stature) and a relatively small %BF was also reflected in the average somatotype of balanced ectomorph (2.4-2.8-3.8). The somatotype observed in the study group was comparable to that found in a previous study on Japanese elite long-distance runners (2.35-2.32-3.82) [9]. However, it is important to acknowledge a wide variability in %BF (8.3–27.7%) between individuals, and therefore caution is required when interpreting the observed mean %BF value. Considering the fact that, like many other Asians, Japanese tend to accumulate a large %BF relative to their BMI [18], the results may indicate the likelihood that some of the Japanese female long-distance runners who participated in the study may carry a large amount of body fat relative to their body size, possibly as subcutaneous fat.

In the current study, although the average BMI of the participants was classified as "underweight" and comparable with that of elite athletes, the participants expressed a strong desire to have a taller and lighter physique that makes their ideal BMI lower. In addition, more than half of the participants perceived themselves as being fat and as having an excessive amount of fat tissues. These results observed from Japanese female university long-distance runners are in line with previous studies, suggesting that female long-distance runners tend to express strong body dissatisfaction [13–15]. However, a number of studies reported strong body dissatisfaction and inaccurate body perceptions among Japanese young females in general [19,21,37], even compared with non-Japanese eating-disordered and overweight females [20]. Compared to previous studies that examined the body dissatisfaction and body concern of young Japanese females using BSS and BAQ [21,23,37], the observed results are consistent. Further, the risk of disordered eating as determined with EAT-26 was also comparable to that of Japanese young females in general [23]. This suggests that the observed results from the participants of the current study were simply due to their ethnic characteristics and not to their participation in a long-distance running event. Kagawa et al. [19] reported that the difference between young Japanese females' measured and perceived ideal weight was approximately 4 kg. In comparison, the participants of the present study showed a difference between their measured and perceived ideal weight ranging between 2.1 and 2.9 kg, depending on the purpose of their ideal (i.e., ideal for current weight, for ideal height or for their competition). From these results, although they show comparable BSS and BAQ results, it can be said that the participants' desire for thinness was rather weak compared with that of young Japanese females. The smaller difference observed between the perceived ideal body mass and the measured values may be simply due to the fact that many participants were lighter than young Japanese females

in general. However, the results may also indicate a possibility that they had a better understanding of their current body mass than non-athletes. This is possible if participants regularly measure their body mass as a part of training. It is also possible that they are sensitive about their body mass and its change if they had experienced a deterioration of their health condition or performance due to excessive weight loss in the past.

It has been well documented that body image is a multi-dimensional concept and is influenced by a number of internal and external factors. While the influences of family, peers and the media (including social media) have been well recognized as the Tripartite Influence Model [38,39], a number of researchers have suggested the importance of pressure from coaches to maintain a low body weight and lean physique as a potential risk factor for body dissatisfaction and the development of eating disorders for athletes [40,41]. Furthermore, a number of studies have indicated that actual physique, not only results of simple indices like BMI but also the amount and accumulation pattern of adipose tissue and bone diameter, influence body dissatisfaction [21,42,43]. In the present study, however, the body dissatisfaction of the participants was not associated with the measured anthropometric or body composition parameters. This suggests that the study participants expressed body dissatisfaction and a desire for thinness that were not based on their current anthropometric or body composition parameters. This also suggests that they have different sources other than their actual physique that trigger their body dissatisfaction. However, while 30–40% of them expressed that the Internet and TV were their sources of information, no other options questioned in the study were found to be their information source regarding their body. These results may suggest that the participants' body dissatisfaction is not based on their current physique or external factors.

While the measured variables were not associated with body dissatisfaction, attitude toward the body as measured with BAQ was found to be associated with some anthropometric parameters. Among the anthropometric parameters associated with BAQ, variables measured from the limbs, including triceps, biceps, front thigh skinfolds and mid-thigh and calf maximum girths, were found to be important to the participants' decision-making. In addition, almost all participants were concerned about their body solely in relation to their performance and the factors of the Tripartite Influence Model (i.e., family, peers and the media), and factors specific to the participants such as coaches do not influence their body concern. This study suggests that the body concern of the participants was not based on external factors but rather on their own performance and how they perceive their own physique themselves.

Together with providing a construction of the participants' perceived body dissatisfaction, this study indicates that the body image of the Japanese female university long-distance runners that participated in the present study was strongly influenced by their internal components and not external information. While not being influenced by the media and peer pressure is beneficial, not obtaining accurate and appropriate information or gaining knowledge may be also problematic. If the participants do not gain the adequate knowledge necessary to maintain and improve their performance but rather build their ideal physique based on their body mass, possibly according to their past experiences including their best time during their high-school period, it is possible that they will undergo weight management such as unnecessary diet restrictions. Such behaviours may result in an insufficient intake of energy and nutrients that can lead to low energy availability (EA) and less of the protein required to gain skeletal muscle to improve their performance. Insufficient dietary intake may also increase the risk of physiological and metabolic abnormalities including hormonal imbalances and deteriorated bone metabolism that may increase the risk of a number of health conditions such as iron-deficient anaemia, menstrual dysfunction, lowered concentration and stress fractures. In order to improve performance, an increase in skeletal muscle mass and cardiorespiratory function is essential for long-distance runners. Considering that there is no solid conclusion about a relationship between low BMI and performance to date [5,6] and the fact that the highest percentage of male long-distance (i.e., 3000 m and marathon) runners has a BMI of 20 kg/m$^2$ [3], some

degree of weight gain should be acceptable as long as it is due to an increase in skeletal muscle mass. The present study was unable to clarify if the participants truly did not obtain any information related to the improvement or modification of their physique from external sources or were not feeling any pressure or stresses. Therefore, further investigation is warranted to confirm the present findings. In addition, this study could not determine if the participants accepted information from external sources such as nutritional seminars to gain correct knowledge or implemented the provided knowledge into their daily activities by replacing their original values related to their own physique that may have been formed from their own experiences.

The present study has a number of limitations. This study was conducted with a relatively small sample size, and participants were recruited from a single university. Consequently, the study results cannot be generalized to a wider population. In addition, this study did not observe performance as well as body mass when the participants were in high school. Therefore, it was unable to examine the influence of past experience on the participants' current body image. In order to better understand body dissatisfaction and body concern among this particular population, it is recommended to conduct further research using a larger sample size recruited from different universities and incorporate additional questions related to their performance and body size during their high-school period. From a detailed anthropometry, the present study shows that the participants had a body mass close to their perceived ideal weight, and their responses to the BAQ were associated with a number of anthropometric parameters, especially those of the limbs. Because this study was unable to clarify the reasons for the observed results, further investigation is recommended. Furthermore, while the present study shows that the participants prioritize their performance as a reason for considering their physique, this study was unable to determine how well they accept knowledge from external sources and incorporate it into their behaviour modifications. In order to better understand the effectiveness of support activities such as nutritional seminars that incorporate both physiological and psychological components for this particular population, intervention studies may be warranted in the future.

## 5. Conclusions

The present study examined the anthropometric and body composition of Japanese female long-distance runners as well as the different aspects of their body image and investigated factors that influence their body image. Although the participants had a relatively low BMI and %BF on average, they expressed a strong desire to lose weight, and about 50–60% of them perceived themselves as being fat or carrying a large amount of %BF. Their body dissatisfaction was not associated with their current physique or external factors. In comparison, body attitudes and concerns were associated with the appearance of their limbs and performance. In order to better understand the factors that influence their body dissatisfaction and the effects of providing accurate information on behaviour modification, further investigation is warranted.

**Author Contributions:** Conceptualization, S.I. and M.K.; methodology, S.I. and M.K.; formal analysis, M.K.; investigation, S.I., M.K., K.I.-T. and M.O.; writing, M.K., S.I., K.I.-T. and M.O.; project administration, S.I.; funding acquisition, S.I. All authors have read and agreed to the published version of the manuscript.

**Funding:** This research was funded by Toyo University Olympic and Paralympic promotion special research project B.

**Institutional Review Board Statement:** The study was conducted in accordance with the Declaration of Helsinki (2013 revised edition, Fortaleza) and was approved by the Ethics Committee of Toyo University (TU2017-010TU2017-H0052018-H002).

**Informed Consent Statement:** Informed consent was obtained from all participants prior to their involvement in the study.

**Data Availability Statement:** The data are not available to the public to ensure the privacy of the participants.

**Acknowledgments:** The authors would like to express their appreciation to all the participants, coaches and support staff of the team for their support of the study. The authors also thank the ISAK-accredited anthropometrists and students who were involved in the data-collection session.

**Conflicts of Interest:** The authors declare no conflict of interest.

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
