# Peer review of "Physical Characteristics and Body Image of Japanese Female University Long-Distance Runners"

_applsci, doi:10.3390/app13116442_

Round 1

Reviewer 1 Report

The study investigated the association between self body image and anthropometry measures of Japanese distance runners. The manuscript is generally well written with detailed introduction and methodology, and clear discussion of results.

I do not see any technical issues with the manuscript. There are only some minor grammar issues that the authors may consider amending.

Minor editing of English language required

Author Response

The study investigated the association between self body image and anthropometry measures of Japanese distance runners. The manuscript is generally well written with detailed introduction and methodology, and clear discussion of results.

I do not see any technical issues with the manuscript. There are only some minor grammar issues that the authors may consider amending.

Thank you very much for insightful comments and suggestions. We have revised the manuscripts and corrected based on your suggestions.

Line 22: “is still limiting” change to “is still limited”

Thank you very much for you suggestion. It has been corrected.

Line 25: “conducted to” change to “conducted on”

Thank you very much for you suggestion. It has been corrected.

Line 26: “includes” change to “included”

Thank you very much for you suggestion. It has been corrected.

Line 30: change the full stop after %BF to a comma

Thank you very much for you suggestion. It has been corrected.

Line 31: Can remove the word “current”

Thank you very much for you suggestion. It has been corrected.

Line 40: “Sports performance is known to be associated with a physique” may consider changing to “Sports performance is known to be associated with specific types of physique”

Thank you very much for you suggestion. It has been corrected.

Line 44: “participate” change to “participating”

Thank you very much for you suggestion. It has been corrected.

Line 46: “particularly of marathon” change to “particularly for marathon”

Thank you very much for you suggestion. It has been corrected.

Line 94: “participating Ekiden” change to “participating in Ekiden “

Thank you very much for you suggestion. It has been corrected.

Line 234: “put effort” change to “put in effort”

Thank you very much for you suggestion. It has been corrected.

Line 236: “and aware of” change to “and are aware of”

Thank you very much for you suggestion. It has been corrected.

Line 308: “While a relatively small sample size may contribute the results” change to “While a relatively small sample size contributed to the results”

Thank you very much for you suggestion. It has been corrected.

Line 329: “interpret” change to “interpreting”

Thank you very much for you suggestion. It has been corrected.

Line 331: “that some of Japanese female long-distance runners participated” change to “that some of the Japanese female long-distance runners who participated”

Thank you very much for you suggestion. It has been corrected.

Line 369: “an” change to “the”

Thank you very much for you suggestion. It has been corrected.

Line 387-390: “In addition ……concern.” This sentence need amendment as it does not seem to be expressing the information well.

Thank you very much for your suggestion. We have revised the sentence to make it more explicit.

Reviewer 2 Report

The aim of this study was to investigate physical characteristic of Japanese female long-distance runners and also examine associations between these physical characteristics and body image.

Authors provide sufficient background and include relevant references. The materials and methods section could be improved. The article must be seen in the light of some limitations that need correction:

Line 182: among others.. fat-free soft tissue mass has no abbreviation (FFSTM)

Table 3. Are result presented as mean+-SD? No information included in the table

Table 4 is not self-explanatory. Abbreviations should be explained below the table.

Figure 1: For what n=22 state for in this figure?

Sample size is was very small to make general statement for the population of female long distance runners. It would be hard to say the results show a trend as the study included participants only from one university.

Author Response

The aim of this study was to investigate physical characteristic of Japanese female long-distance runners and also examine associations between these physical characteristics and body image.

Authors provide sufficient background and include relevant references. The materials and methods section could be improved. The article must be seen in the light of some limitations that need correction:

Thank you very much for your time to review our manuscript and providing us valuable comments. We have considered your comments and revised the manuscripts.

Line 182: among others. fat-free soft tissue mass has no abbreviation (FFSTM).

Thank you very much for your suggestion. Since the term “fat-free soft tissue mass” appears in the text only once, we have decided there is no need to abbreviate this term for this manuscript.

Table 3. Are result presented as mean+-SD? No information included in the table.

Thank you very much for your comment. As stated in line 252 – 254, the results were presented as mean ± standard deviation or (median; quartiles) or percentage (%).

Table 4 is not self-explanatory. Abbreviations should be explained below the table.

Thank you very much for your suggestion. We have included an explanation of the table and abbreviations below the table.

Figure 1: For what n=22 state for in this figure?

Thank you very much for a comment. Since the number of participants analyzed in the present study varies by the questions, we have stated the number of participants used in the analysis shown in Figure 1 and Figure 2.

Sample size is was very small to make general statement for the population of female long distance runners. It would be hard to say the results show a trend as the study included participants only from one university.

Thank you very much for your comment. We agree with the reviewer’s opinion and therefore we have stated this as one of the limitations in line 432 – 434.

Reviewer 3 Report

The article is appropriate for the character and application of the Journal. The design of the sequence of the contents is appropriate and puts in knowledge a problem/issue that concerns coaches and athletic trainers.

The introduction to the article finally links with the objectives, but the second objective is very open and ambitious, and in reference to the information provided in the rest of the article, the second objective should be reworded so that it agrees with the results and conclusions of the article.

Regarding the materials and methods, the study sample is scarce and this deficit is recognized by the authors, but the specialization of the sample and its difficult access to scientific research, it is important to indicate as the authors say that only one university center has been analyzed, which further limits the study.

As for the measurement tools, they are correct and well-founded. The questionnaire (validated/non-validated) of which the results are presented in relation to "major sources of information on physique and body composition were examined" does not appear in LINEA 284 It would be important to have more details of this process or to evaluate its omission.

In the Discussion there are reflections that are not argued or are difficult to demonstrate.

Line 360 and 420, these paragraphs do not belong to the discussion, perhaps they are statements in line with the final recommendations close to the conclusion.

Line 385. This statement is not supported by any scientific citation and lacks the force of argument, either it is reformulated with scientific support or it is excluded.

The recommended bibliography is correct.

Author Response

The article is appropriate for the character and application of the Journal. The design of the sequence of the contents is appropriate and puts in knowledge a problem/issue that concerns coaches and athletic trainers.

Thank you very much for your time to review our manuscript and providing us valuable comments. We have revised the manuscripts based on your suggestions.

The introduction to the article finally links with the objectives, but the second objective is very open and ambitious, and in reference to the information provided in the rest of the article, the second objective should be reworded so that it agrees with the results and conclusions of the article.

Thank you very much for your suggestion. We have revised the second objective and split it into two objectives of 2) to explore body image of the study participants, and to 3) determine factors that influence body image in order to better reflect the results. The conclusion was also revised accordingly.

Regarding the materials and methods, the study sample is scarce and this deficit is recognized by the authors, but the specialization of the sample and its difficult access to scientific research, it is important to indicate as the authors say that only one university center has been analyzed, which further limits the study.

Thank you very much for your suggestion. Limitations about a small sample size and insufficiency to generalize the results have been stated in Line 435-436. In order to improve generalizability of the study, we have added a need of a larger sample size recruited from different universities in future research.

As for the measurement tools, they are correct and well-founded. The questionnaire (validated/non-validated) of which the results are presented in relation to "major sources of information on physique and body composition were examined" does not appear in LINEA 284 It would be important to have more details of this process or to evaluate its omission.

Thank you very much for pointing out missing description for the question about major sources of information that influence their ideal physique and body composition. Together with description for the question on reasons to concern their own physique, we have added description about this question in the methods section.

Line 360 and 420, these paragraphs do not belong to the discussion, perhaps they are statements in line with the final recommendations close to the conclusion.

Thank you very much for your suggestion. We have moved these paragraphs to future recommendations close to the conclusion.

Line 385. This statement is not supported by any scientific citation and lacks the force of argument, either it is reformulated with scientific support or it is excluded.

Thank you very much for your comment. Since this sentence is based on subjective impression of the authors and unable to provide references for scientific evidence, we have removed the sentence. Instead, we have added a sentence to emphasize a need of future research to explore reasons for associations between BAQ and anthropometric variables of the limbs.

Round 2

Reviewer 2 Report

Thank you for the corrections that have been made.

Reviewer 3 Report

Authors have done changes and added required information for aim goal the scienst quality of the review. Acepted.